# ESTIMATING POST-SYNAPTIC EFFECTS FOR ONLINE TRAINING OF FEED-FORWARD SNNS

## ABSTRACT

Facilitating online learning in spiking neural networks (SNNs) is a key step in developing event-based models that can adapt to changing environments and learn from continuous data streams in real-time. Although forward-mode differentiation enables online learning, its computational requirements restrict scalability. This is typically addressed through approximations that limit learning in deep models. In this study, we propose Online Training with Postsynaptic Estimates (OTPE) for training feed-forward SNNs, which approximates Real-Time Recurrent Learning (RTRL) by incorporating temporal dynamics not captured by current approximations, such as Online Training Through Time (OTTT) and Online Spatio-Temporal Learning (OSTL). We show improved scaling for multi-layer networks using a novel approximation of temporal effects on the subsequent layer's activity. This approximation incurs minimal overhead in the time and space complexity compared to similar algorithms, and the calculation of temporal effects remains local to each layer. We characterize the learning performance of our proposed algorithms on multiple SNN model configurations for rate-based and time-based encoding. OTPE exhibits the highest directional alignment to exact gradients, calculated with backpropagation through time (BPTT), in deep networks and, on time-based encoding, outperforms other approximate methods. We also observe sizeable gains in average performance over similar algorithms in offline training of Spiking Heidelberg Digits with equivalent hyper-parameters (OTTT/OSTL – 70.5%; OTPE – 75.2%; BPTT – 78.1%).

## 1 INTRODUCTION

Spiking neural networks (SNNs) promise a path toward energy-efficient machine intelligence for streaming data (Yik et al., 2023). Despite this potential, efficiently training them on temporal sequences remains a challenge. Backpropagation through time (BPTT) applied with surrogate gradient estimates of SNN neurons (Neftci et al., 2019; Zenke and Ganguli, 2018; Shrestha and Orchard, 2018) has become the dominant method to train SNNs. However, BPTT is unsuitable for online learning (Kaiser et al., 2020; Bohnstingl et al., 2022; Rostami et al., 2022)

Real-time recurrent learning (RTRL) (Williams and Zipser, 1989) computes exact gradients for stateful models without temporal unrolling. This enables online learning at the cost of $O(n^3)$ storage and $O(n^4)$ compute, which is not practical for training all but the smallest networks. To address these limitations, practical implementations approximate RTRL by adopting low-rank matrix approximations (Mujika et al., 2018; Benzing et al., 2019), leveraging stochastic gradient estimates (Tallec and Ollivier, 2018), incorporating model-specific assumptions (Bohnstingl et al., 2022), or selectively omitting certain influence pathways (Menick et al., 2020). Of these, a promising approach for SNN training involves storing only the influence matrix to gradient values along the diagonal of the stored Jacobians. This approximation restricts temporal influence of the gradient calculations to the current output from a single layer, which neglects how the previous outputs of a neuron impact the membrane potential of downstream neurons. This approximation reduces learning performance when compared to the exact gradient calculations in RTRL or BPTT (Bohnstingl et al., 2022), due to untracked causal effects over time. To address this, we develop a novel approximation of this temporal effect through our algorithm, Online Training with Postsynaptic Estimates (OTPE). OTPE maintains a trace of parameter influence over multiple time-steps, implementing a comprehensive approximation of the entire temporal effect in a one-hidden-layer model. In deep networks, spikes

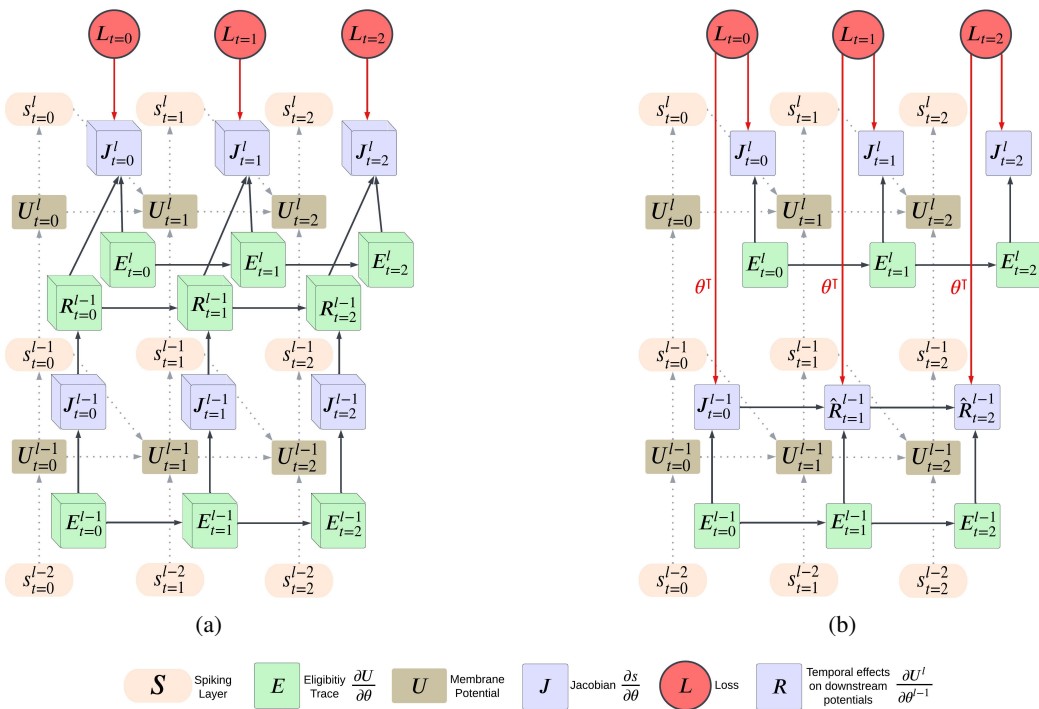

Figure 1: Depiction of RTRL and OTPE. The grey dotted lines indicate the forward pass effects through a feed-forward spiking neural network. (a) A graphical depiction of computations incurred in RTRL. Here, the Jacobian ($J$) matrix is calculated for all the upstream parameters in the model. Thus, we represent this by gradients in cubes, reflecting the $n^3$ size of the Jacobian. (b) Depiction of OTPE. Unlike RTRL, all temporal gradient calculations are layer-local, and loss is backpropagated.

generated in earlier layers will have a delayed influence on spiking activity of neurons in deeper layers. Thus, gradient approximation worsens as the error is back-propagated if only the current time-step is considered. Intuitively, the performance difference between Online Spatio-Temporal Learning (OSTL) Bohnstingl et al. (2022) and BPTT reflects the impact of these residual temporal effects since the explicit difference between OSTL and exact gradient calculation is the exclusion of these residual effects. We demonstrate that OTPE's approximation significantly increases alignment with exact gradients. We observe a $\sim 70\%$ increase in gradient cosine similarity to BPTT, in the first hidden layer and $\sim 50\%$ in the second hidden layer of a 2-hidden-layer network for a model trained on the Spiking Heideberg Digits (SHD) dataset (Cramer et al., 2020). We consistently observe similar improvements for both online and offline learning, and across other evaluated datasets and model configurations.

Our primary contributions include:

- a novel approximation of RTRL, OTPE, that captures the effects of multi-step temporal sequences through a spiking neuron which are excluded from previous algorithms;
- a further relaxation of our algorithm that can achieve similar scalability to state-of-the-art while delivering superior learning performance on multiple tasks;
- in-depth evaluations of OTPE against existing SNN training algorithms, including gradient approximation quality and learning performance in temporal and rate-encoded inputs.

## 2 BACKGROUND

Training stateful networks such as SNNs requires temporal credit assignment (Maeda and Wakamura, 2005; Lillicrap et al., 2016; Bohnstingl et al., 2022; Xiao et al., 2022; Kaiser et al., 2020). Of these different methods, state-of-the-art results in SNN training typically employ BPTT-based credit assignment (Eshraghian et al., 2023) with surrogate derivatives to calculate gradients through the

SNNs discontinuity (Zenke and Vogels, 2021; Zenke and Ganguli, 2018). Consequently, we compare the gradient approximation quality of OTPE, Online Training Through Time (OTTT)(Xiao et al., 2022), and OSTL against the exact gradients computed by BPTT. All methods are applied to deep feed-forward SNNs composed of leaky integrate-and-fire (LIF) neurons in fully connected layers.

## 2.1 LIF NEURON

SNNs promise low computational requirements, arising from activation sparsity and unary outputs. LIF neurons are the most commonly used for balancing performance and complexity, especially in deep models (Zenke and Ganguli, 2018). Similar to a plethora of other work (Fang et al., 2021), we use a subtraction-based reset formulation of the LIF, with the neuron reset behavior written as

$$Reset: \quad U_t^l = \lambda U_{t-1}^l - V_{th} \cdot s_t^l, \quad s_t^l = H(\lambda U_{t-1}^l + s_t^{l-1} \cdot \theta - V_{th}).$$

Here, $U_t^l$ is the neuron's membrane potential in layer $l$ at time-step $t$, which decays by the leak $\lambda$ while accumulating spiking inputs $s_t^{l-1}$ from the previous layer, weighted by $\theta$. The neuron emits a spike $s_t^l$ whenever its membrane potential exceeds the threshold $V_{th}$. The derivative of the Heaviside step function ($H$) is the Dirac delta function which is zero almost everywhere, effectively setting all gradients to zero. To generate non-zero gradients, we employ surrogate gradients, which replace the dirac delta function with the derivative of the fast sigmoid function (Neftci et al., 2019).

## 2.2 RTRL

Real-Time Recurrent Learning (RTRL) (Fig. 1) calculates gradients through time using forward-mode differentiation, calculating and storing Jacobian-vector products. While BPTT must store outputs at each layer and unroll the network to perform reverse-mode differentiation through time, RTRL stores and updates each parameter's effects on the network's state. Because the stored Jacobian tracks every parameter's influence on each state variable ($O(n^3)$ in the number of parameters) the network avoids unrolling to calculate gradients. Due to this, RTRL can calculate exact gradients for online learning. RTRL gradient calculation for the output layer of an SNN can be written as

$$
\begin{aligned}
\frac{\partial \mathcal{L}}{\partial \theta^l} &= \sum_{t=1}^{T} \frac{\partial \mathcal{L}_t^l}{\partial s_t^l} \frac{\partial s_t^l}{\partial U_t^l} \frac{\partial U_t^l}{\partial \theta^l} \\
&= \sum_{t=1}^{T} \frac{\partial \mathcal{L}_t}{\partial s_t^l} \frac{\partial s_t^l}{\partial U_t^l} \left( \frac{\partial U_t^l}{\partial \theta_t^l} + \frac{\partial U_t^l}{\partial U_{t-1}^l} \frac{\partial U_{t-1}^l}{\partial \theta^l} \right).
\end{aligned}
\tag{1}
$$

We denote the loss with $\mathcal{L}$, the spike output with $s$, the membrane potential with $U$, the parameters with $\theta$, the total number of time-steps with $T$, and the current time-step with $t$. We can recursively calculate and store the temporal gradients, $\frac{\partial U_t^l}{\partial \theta^l}$, in eqn (1) through $\frac{\partial U_t^l}{\partial \theta^l} = \left( \frac{\partial U_t^l}{\partial \theta_t^l} + \frac{\partial U_t^l}{\partial U_{t-1}^l} \frac{\partial U_{t-1}^l}{\partial \theta^l} \right)$.

For the hidden layer, we can expand eqn (1), substituting $\frac{\partial U_t^l}{\partial \theta^l}$ with $\frac{\partial U_t^l}{\partial \theta_t^{l-1}}$, resulting in

$$
\begin{aligned}
\frac{\partial U_t^l}{\partial \theta_t^{l-1}} &= \frac{\partial U_t^l}{\partial s_t^{l-1}} \frac{\partial s_t^{l-1}}{\partial \theta_t^{l-1}} \\
&= \frac{\partial U_t^l}{\partial s_t^{l-1}} \left( \frac{\partial s_t^{l-1}}{\partial U_t^{l-1}} \left( \frac{\partial U_t^{l-1}}{\partial \theta_t^{l-1}} + \frac{\partial U_t^{l-1}}{\partial U_{t-1}^{l-1}} \frac{\partial U_{t-1}^{l-1}}{\partial \theta^{l-1}} \right) \right).
\end{aligned}
\tag{2}
$$

Where, the kernel's transpose in a dense layer, $\theta^\intercal$, is given by $\frac{\partial U_t^l}{\partial s_t^{l-1}}$.

## 2.3 PRACTICAL APPROACHES TO RTRL

Prior work achieves online capabilities by approximating exact gradient computation. Of particular interest are OSTL (Bohnstingl et al., 2022) and OTTT (Xiao et al., 2022). OSTL focuses on achieving bio-plausibility by implementing eligibility traces (Gerstner et al., 2018), through a mechanism derived from RTRL. Another similar algorithm is OTTT, which ignores the reset mechanism in the gradient calculation of SNNs to reduce the storage and compute overhead relative to OSTL. See Table 1 for the time and space complexity of the evaluated algorithms.

### 2.3.1 OSTL

OSTL is an RTRL approximation that separates spatial and temporal gradients for calculating the overall gradients. Because $\frac{\partial U_t}{\partial \theta}$ in RTRL has $n^3$ entries (where n is the layer size), the storage demand of RTRL limits scalability. OSTL approximates $\frac{\partial U_t}{\partial \theta}$ by assuming that all nonzero elements are along the diagonal, thus reducing its size to $n^2$. This assumption holds for a feed-forward SNN, achieving exact gradient computation in a network without hidden layers.

In order to train a deep network, OSTL backpropagates a spatial gradient through the current time-step ($t$) and combines this with a temporal gradient, which maintains how a layer's parameters influenced its most recent output, $s_t$. While all temporal dynamics concerning a single layer's influence on $s_t$ are accounted for, the influence of any spiking activity at previous time-steps is excluded, as shown

$$\frac{\partial \mathcal{L}}{\partial \theta^{l-1}} = \sum_{t=1}^{T} \frac{\partial \mathcal{L}_t^l}{\partial s_t^l} \frac{\partial s_t^l}{\partial U_t^l} \frac{\partial U_t^l}{\partial \theta^{l-1}} = \sum_{t=1}^{T} \frac{\partial \mathcal{L}_t}{\partial s_t^l} \frac{\partial s_t^l}{\partial U_t^l} \left( \frac{\partial U_t^l}{\partial \theta_t^{l-1}} + \cancel{\frac{\partial U_t^l}{\partial U_{t-1}^l} \frac{\partial U_{t-1}^l}{\partial \theta^{l-1}}} \right).$$

### 2.3.2 OTTT

OTTT is conceptually similar to OSTL. Since $\frac{\partial s_t^l}{\partial U_t^l}$ is the Dirac delta function, a surrogate derivative normally takes its place, but OTTT chooses only to apply the surrogate derivative when calculating the spatial gradient and refrains from doing so when updating the temporal gradient. In other words, the gradient calcuation from BPTT which uses $s_t^{l-1}$ can be substituted in for the calculation in RTRL for $\frac{\partial U_t^l}{\partial \theta_t^l}$. Since the derivative of the Heaviside step function is zero almost everywhere, the temporal gradient in a subtraction-based LIF neuron simplifies to the summation over time of the product between the time-weighted leak $\lambda^{T-t}$ and $s_t^{l-1}$. Consequently, only a running weighted sum of the input, $\hat{a}$ in Xiao et al. (2022), is required to be stored and updated, reducing the space complexity of OTTT to $O(n)$. While the surrogate derivative is normally applied as $\frac{\partial U_t}{\partial U_{t-1}} = \lambda + \frac{\partial U_t}{\partial s_t} \frac{\partial s_t}{\partial U_{t-1}}$. OTTT uses the Heaviside function instead, such that $\frac{\partial U_t}{\partial U_{t-1}} = \lambda + \frac{\partial U_t}{\partial s_t} \cdot \emptyset$, which is simply $\lambda$. Consequently, OTTT's temporal gradient, $\hat{a}$, can be calculated as $\hat{a}_{t=T}^l = \sum_{t=1}^{T} \lambda^{T-t} s_t^{l-1}$.

## 3 POST-SYNAPTIC ESTIMATION FOR SNNS

In a feed-forward SNN with hidden layers, temporal dynamics that influence subsequent layers in the network are not addressed by current approximations. We achieve a space-efficient method of approximating these gradients in a similar fashion to OTTT.

Specifically, we do not apply the surrogate derivative for $\frac{\partial s_{t+1}^{l+1}}{\partial U_t^{l+1}}$ when calculating $\frac{\partial U_{t+1}^{l+1}}{\partial U_t^{l+1}}$ during the calculation of the temporal dynamics in layer $l$. We also assume the time constant is global. During forward gradient computation in layer $l$, we assume the subsequent layer's temporal dynamics are captured by the running sum of the subsequent layer's inputs, which are the output spikes of layer $l$. By maintaining a running weighted sum of how the parameters in layer $l$ influenced previous spikes, the parameters' contribution to the subsequent layer's temporal dynamics is represented during gradient calculation. In order to calculate how the parameters affect spikes at the current time-step, OTPE implements OSTL to produce the diagonal Jacobian matrix, $\frac{\partial s_t^l}{\partial \theta^l}$ at each time-step.

OTPE selectively applies the surrogate derivatives such that

$$\frac{\partial \mathcal{L}}{\partial \theta^{l-1}} = \sum_{t=1}^{T} \frac{\partial \mathcal{L}_t^l}{\partial s_t^l} \frac{\partial s_t^l}{\partial U_t^l} \frac{\partial U_t^l}{\partial \theta^{l-1}} = \sum_{t=1}^{T} \frac{\partial \mathcal{L}_t}{\partial s_t^l} \frac{\partial s_t^l}{\partial U_t^l} \left( \frac{\partial U_t^l}{\partial \theta_t^{l-1}} + \lambda \cdot \frac{\partial U_{t-1}^l}{\partial \theta^{l-1}} \right). \tag{3}$$

Since $\frac{\partial U_t^l}{\partial s_t^{l-1}}$ in eqn (2) is the kernel's transpose, $\theta^\intercal$, we avoid recursion, and can write

$$\frac{\partial U_{t=T}^l}{\partial \theta^{l-1}} = \theta^{l\intercal} \sum_{t=1}^{T} \lambda^{T-t} \frac{\partial s_t^{l-1}}{\partial \theta^{l-1}} = \theta^{l\intercal} \hat{R}^{l-1}. \tag{4}$$

| Name | Space Complexity | Time Complexity |
|------|:----------------:|:---------------:|
| BPTT | $O(Tn)$ | $O(Tn^2)$ |
| OTTT | $O(n)$ | $O(Tn + n^2)$ |
| OSTL | $O(n^2)$ | $O(Tn^2)$ |
| Approx OTPE | $O(n)$ | $O(Tn + n^2)$ |
| OTPE | $O(n^2)$ | $O(Tn^2)$ |

Table 1: Time and space complexity of BPTT and all tested approximate algorithms for calculating gradients at time-step $T$. The complexity is in reference to a single dense layer with an input and output size $n$ and batch size of 1.

As seen in eqn (4), similar to OSTL, we only need to maintain a running weighted sum of $\frac{\partial s_t^{l-1}}{\partial \theta^{l-1}}$, which we denote as $\hat{R}$. OTPE differs from OSTL by approximating the term OSTL ignores, which is the temporal gradient from post-synaptic neurons. Q3: Because $\hat{R}_{t=0} = \frac{\partial U_{t=0}^l}{\partial \theta_{t=0}^{l-1}}$ and updates as $\hat{R}_t = \frac{\partial U_t^l}{\partial \theta_t^{l-1}} + \lambda \cdot \hat{R}_{t-1}$, $\hat{R}$ is the same shape as $\frac{\partial U_t^l}{\partial \theta_t^{l-1}}$, which is $n^2$ due to OSTL's sparse assumption.

$$\frac{\cancel{\partial U_t^l}}{\cancel{\partial U_{t-1}^l}} \frac{\cancel{\partial U_{t-1}^l}}{\cancel{\partial \theta^{l-1}}} \approx \lambda \cdot \theta^{l\mathsf{T}} \hat{R}^{l-1} , \quad \frac{\partial \mathcal{L}}{\partial \theta^{l-1}} \approx \sum_{t=1}^{T} \frac{\partial \mathcal{L}_t}{\partial s_t^l} \frac{\partial s_t^l}{\partial U_t^l} \left( \frac{\partial U_t^l}{\partial \theta_t^{l-1}} + \lambda \cdot \theta^{l\mathsf{T}} \hat{R}^{l-1} \right) .$$

### 3.1 APPROXIMATE OTPE

Although OTPE achieves scalability comparable to OSTL, OTTT remains more scalable. To combine the benefits of OTPE's temporal information with $O(n)$ space complexity, we make two assumptions on top of OTPE: 1) We decouple the components of $\hat{R}$, $\frac{\partial s_t^l}{\partial U_t^l}$ and $\frac{\partial U_t^l}{\partial \theta^l}$, through time by assuming $\sum_{t=1}^{T} \lambda^{T-t} \frac{\partial s_t^l}{\partial U_t^l} \frac{\partial U_t^l}{\partial \theta^l} \approx \left( \frac{1}{T} \sum_{t=1}^{T} \lambda^{T-t} \frac{\partial s_t^l}{\partial U_t^l} \right) \cdot \left( \sum_{t=1}^{T} \frac{\partial U_t^l}{\partial \theta^l} \right)$. We maintain a size $n$ running weighted average of the surrogate gradients through time, which we refer to as $\bar{g} = \frac{1}{T} \sum_{t=1}^{T} \lambda^{T-t} \frac{\partial s_t^l}{\partial U_t^l}$. 2) We approximate our layer-local temporal calculation in OTPE to only store vectors of size $n$ by assuming the same temporal dynamics as OTTT for a single layer, $\frac{\partial U_t^l}{\partial \theta^l} \approx \hat{a}$ , such that $\frac{\partial U_t^l}{\partial \theta^{l-1}} \approx \theta^{l\mathsf{T}} \cdot \bar{g} \left( \hat{a}_t^{l-1} + \lambda \cdot \hat{z}_{t-1}^{l-1} \right)$, where $\hat{z}$ is a weighted sum of OTTT's weighted sum ($\hat{z}_t^{l-1} = \hat{a}_t^{l-1} + \lambda \cdot \hat{z}_{t-1}^{l-1}$). When the spatial gradient reaches a layer, $\bar{g}$ is used in place of the immediate time-step's surrogate. The kernel gradients are calculated by taking the outer product of the back-propagated loss and $\hat{z}$. When considering networks with multiple hidden layers, backpropagating error for OTPE or Approx OTPE is similar to OSTL and OTTT, with the exception of using $\bar{g}$ instead of the most recent time-step's surrogate gradients.

$$\frac{\partial \mathcal{L}_T^l}{\partial \theta^{l-1}} \approx \left( \left( \frac{\partial \mathcal{L}_T^l}{\partial s_T^l} \frac{\partial s_T^l}{U_T^l} \cdot \theta^{l\mathsf{T}} \right) \cdot \bar{g} \right) \otimes \hat{z} \tag{5}$$

### 3.2 F-OTPE

OTPE calculates gradients under the assumption that output spikes of a hidden layer at previous time-steps are accumulated with membrane leak in the following layer ($\hat{R}^l = \frac{\partial \sum_{t=1}^{T} \lambda^{T-t} s_t^l}{\partial \theta^l}$). In the output layer, this can apply to the leaked accumulation of spikes through time. Suppose we calculate loss using the accumulated spikes instead of solely relying on the spiking behavior at the current time-step. In this case, we can use $\hat{R}$ to exactly calculate the derivative of the loss with respect to the output layer's parameters ($\frac{\partial \mathcal{L}}{\partial \sum_{t=1}^{T} \lambda^{T-t} s_t^l} \frac{\partial \sum_{t=1}^{T} \lambda^{T-t} s_t^l}{\partial \theta^l} = \frac{\partial \mathcal{L}}{\partial \sum_{t=1}^{T} \lambda^{T-t} s_t^l} \hat{R}^l = \frac{\partial \mathcal{L}}{\partial \theta^l}$). We test the application of OTPE to all layers in the network for online learning, applying softmax cross-entropy loss to a leaking sum of the model's output. We also do this for its approximation, denoted by "F-".

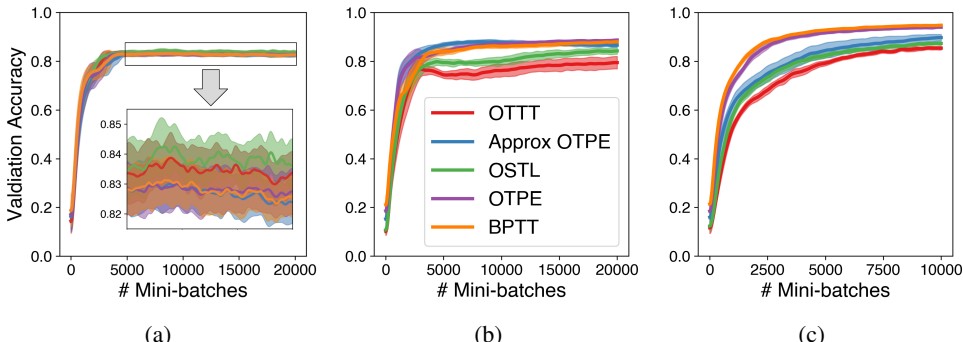

Figure 2: Mean validation accuracy across four seeds throughout offline training for (a) R-Randman, (b) T-Randman, and (c) SHD respectively. Shaded regions indicate one standard deviation.

## 4 EXPERIMENTS

We test and compare all algorithms for online and offline training on Randman (Zenke and Vogels, 2021), a synthetic spiking dataset, and SHD (Cramer et al., 2020) (dataset related parameters provided in A.1). In offline learning, we evaluate accuracy, cosine similarity to exact gradients produced via BPTT, and training trajectories in the loss landscape. In offline training on SHD, we evaluate test accuracy across multiple model configurations to identify suitable hyperparameters for comparing all algorithms. For online training on SHD, we run a learning rate search across three model configurations for the same purpose (see Appendix A.2 for full hyperparameter search results). We exclude f-OTPE from offline learning comparison because loss is calculated differently, preventing a fair comparison of gradient approximation quality. All tests were performed on NVIDIA GPUs, using the Adamax optimizer (Kingma and Ba, 2014). We chose the Adamax optimizer because of its relative effectiveness on temporal learning and use in the original SHD study. We provide JAX code to enable the reproduction of our results (Bradbury et al., 2018; Heek et al., 2023). For all tests, we use a fast sigmoid slope of 25, the default value in snnTorch (Eshraghian et al., 2023). BPTT also delivered highest accuracy with this value in our hyperparameter search (Appendix A.2)

**Randman** is a synthetic dataset, aimed at studying SNN capabilities in learning spike timing patterns. Spike-times are generated on a random smooth manifold, projected onto a high-dimensional hypercube. These are then sampled to generate a spike-train, which the SNN learns to classify. The dimensionality and smoothness of the manifold is varied to adjust the task difficulty. Neurons, firing once per trial, only contain temporal information, a format we term T-Randman. We also modify Randman to test rate-based learning through R-Randman. Here, spike-rates are determined by manifold values and are generated to be temporally uncorrelated through random shuffling.

**Spiking Heidelberg Digits (SHD)** is a spiking dataset consisting of 10,000 recordings of spoken digits 0-9 in German and English, across 20 classes. The recordings are processed through a model of the inner ear to produce a spiking representation. Since Randman's structure does not reflect natural data, we also evaluate model performance on SHD to reflect performance for a practical application. We evaluate accuracy after both online and offline training for SHD.

### 4.1 EVALUATING LEARNING PERFORMANCE

Figure 2 compares OTPE and Approx OTPE against BPTT, OTTT, and OSTL for offline training. We compare performance across multiple datasets (R-Randman, T-Randman, SHD). We train a 2-hidden-layer model with a layer width of 128 for both variations of Randman. Performance on R-Randman is similar across methods (the range of their means is $1.3\%$), and both OSTL and OTTT beat BPTT, OTPE, and Approx-OTPE. However, when evaluated on T-Randman, OTPE and Approx-OTPE accuracy is on-par with BPTT (OTPE's smoothed validation accuracy on SHD is $\sim 0.8\%$ lower, $\sim 0.7\%$ higher on T-Randman, and $\sim 0.1\%$ higher on R-Randman) while OSTL ($\sim 7.4\%$ lower on SHD, $\sim 3.8\%$ lower on T-Randman, and $\sim 1.1\%$ higher on R-Randman) and OTTT ($\sim 9.3\%$

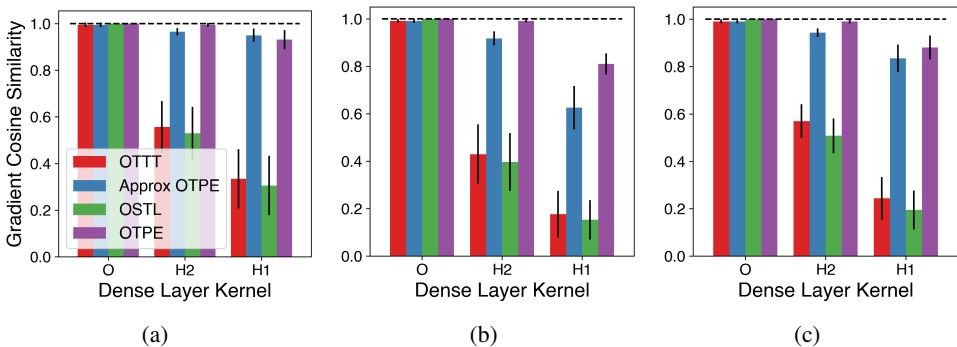

Figure 3: Mean gradient cosine similarity throughout training for each layer, evaluated on (a) R-Randman, (b) T-Randman, and (c) SHD. The output layer (O), has a higher cosine similarity than the earlier layers hidden-1 (H1) and -2 (H2) due to accumulating approximation error during backprop.

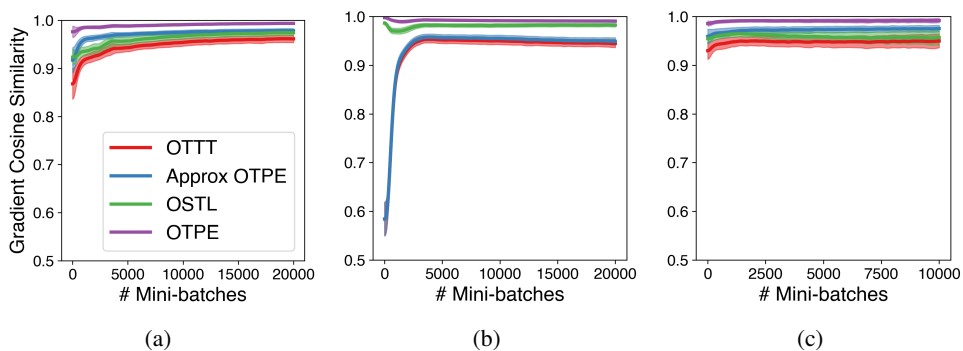

Figure 4: Model-wide, mean gradient cosine similarity over training duration for (a) R-Randman, (b) T-Randman, and (c) SHD. Shaded regions indicate one standard deviation.

lower on SHD, $\sim 8.6\%$ lower on T-Randman, and $\sim 0.7\%$ higher on R-Randman) underperform. All reported numbers are averages over multiple seeds, with SHD results also reported for the last 1000 minibatches. On SHD, Approx OTPE achieves $5.2\%$ lower validation accuracy than BPTT. This is $\sim 2.2\%$ higher than OSTL's validation accuracy and $\sim 4\%$ higher than OTTT. We provide training loss in A.2 and A.3.

Figure 3 shows layer-wise gradient cosine similarity between the gradient estimates of each algorithm (OTPE, Approx OTPE, OTTT, and OSTL), and BPTT for offline learning. We evaluate similarity across the three datasets for 2-hidden-layer networks with 128 layer width in both Randman configurations and a 2-hidden-layer network with 512 layer width. BPTT's gradients are identical to the gradients in the output layers of OSTL and OTPE due to their exact formulation. OTTT and Approximate OTPE are achieve an average $\geq 0.99$ cosine similarity in the output layer. As expected, the gradient similarity with BPTT decreases for deeper layers across all algorithms. OSTL and OTTT incur a $\geq 40\%$ reduction in alignment with BPTT in the second hidden layer (H2), which further decreases to $60\%$ in the first hidden layer (H1). However, OTPE and Approx OTPE are consistently better aligned with those generated by BPTT, with Approx OTPE above 0.6 in H1 for T-Randman compared to $\leq 0.2$ for OSTL and OTTT. We consistently observe these trends in layerwise gradient alignment across the evaluated datasets. Additionally, when observed over the training duration, we see consistent results for layerwise gradient-similarity. The output layers for OTTT and OTPE show improved gradient alignment with BPTT over time ( see Appendix A.3). OTPE shows this increasing alignment with BPTT's gradient directions for the last hidden layer, whereas this trend reverses for OTTT. These trends are consistent across seeds, which may allow the slope of the trends to influence the standard deviation reported in Fig. 3.

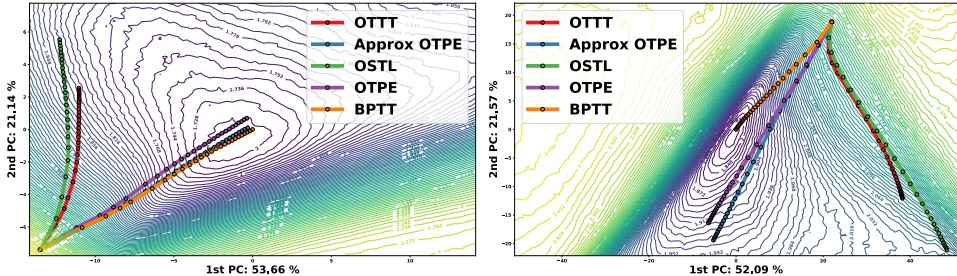

Figure 5: Evaluations of offline learning through the loss landscapes of the different algorithms for (a) R-Randman and (b) T-Randman, evaluated over the validation set. Evaluations are conducted every 800 mini-batches of training with BPTT's model. We observe high similarity between BPTT's model and those trained by OTPE and Approx OTPE in the model weight-space.

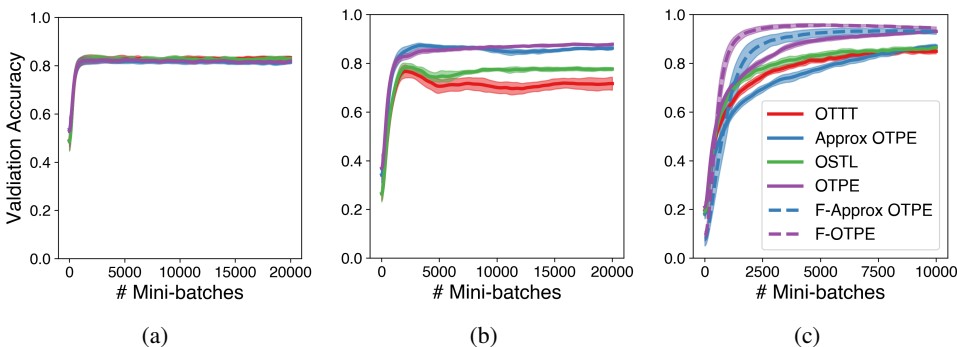

Figure 6: Mean validation accuracy across four seeds for online training evaluated on (a) R-Randman, (b) T-Randman, and (c) SHD. Shaded regions indicate one standard deviation across seeds.

As shown in Fig 4, OTPE consistently achieves the highest model-wise gradient cosine similarity with BPTT over the training duration. When evaluated on T-Randman, OTTT and Approximate OTPE are less aligned with BPTT early in the training (0.6 initially) before stabilizing above 0.9. For SHD and R-Randman, OTPE and its approximation consistently achieve a higher alignment (average of 0.99 and 0.97, respectively) than OSTL and OTTT (average of 0.96 and 0.95) .

We visualize the training loss landscape for different algorithms on the R-Randman and T-Randman tasks (see Figure 5 (a) and (b)), as described in Li et al. (2018). The loss contours are determined by picking a centre point ($\theta*$) and two direction vectors (the initial ($\delta$) and final ($\nu$) BPTT model) and then plotting $f(\alpha, \beta) = L(\theta* +\alpha\delta + \beta\nu)$. The trajectories are rendered by mapping a model, every 800 minibatches during training, into the loss landscape. The strong alignment between BPTT, OTPE, and Approx. OTPE for R-Randman indicates a remarkable similarity between the trained models. For T-Randman, the models trained using OTPE and Approx. OTPE remain more closely aligned to BPTT compared to those trained using OSTL and OTTT. Across these different datasets, the training trajectories of OSTL and OTTT diverge the most from the other models, indicating a significantly altered trained model configuration compared to BPTT.

Table 2 summarizes our performance results for SHD evaluated with different model configurations. We found that OTPE and its approximation perform best with five layers while OTTT, OSTL, and BPTT perform best with three layers. The performance difference between the three-layer and five-layer models is within one standard deviation for BPTT. In contrast OTTT and OSTL incur a substantial drop of 5% and 3.4% in test accuracy after online training, respectively, when training a five-layer model. BPTT unsurprisingly dominates test performance on SHD, but OTPE consistently outperforms OTTT and OSTL. We additionally observe high performance for OTPE across different model and training hyperparameters (data in Appendix A.2). This finding is consistent with Bauer

| Name | Depth | Width | Offline Acc $\pm\sigma$ | Online Acc $\pm\sigma$ |
|---|---|---|---|---|
| OTTT | 3 | 128 | $64.3\% \pm 1.0$ | $66.7\% \pm 1.1$ |
| OSTL | 3 | 128 | $66.7\% \pm 1.0$ | $68.5\% \pm 1.9$ |
| BPTT | 3 | 128 | $\mathbf{73.9}\% \pm 2.5$ | N/A |
| OTPE | 3 | 128 | $72.5\% \pm 0.7$ | $\mathbf{73.6}\% \pm 1.4$ |
| A. OTPE | 3 | 128 | $67.0\% \pm 1.0$ | $69.8\% \pm 1.2$ |
| F-OTPE | 3 | 128 | N/A | $73.3\% \pm 0.9$ |
| F-A. OTPE | 3 | 128 | N/A | $71.2\% \pm 1.7$ |
| OTTT | 3 | 512 | $70.5\% \pm 1.5$ | $71.2\% \pm 0.8$ |
| OSTL | 3 | 512 | $70.5\% \pm 1.5$ | $70.6\% \pm 0.7$ |
| BPTT | 3 | 512 | $\mathbf{78.1}\% \pm 1.0$ | N/A |
| OTPE | 3 | 512 | $75.2\% \pm 0.5$ | $\mathbf{75.4}\% \pm 0.5$ |
| A. OTPE | 3 | 512 | $71.2\% \pm 1.1$ | $71.8\% \pm 1.1$ |
| F-OTPE | 3 | 512 | N/A | $75.3\% \pm 0.3$ |
| F-A. OTPE | 3 | 512 | N/A | $71.5\% \pm 1.2$ |
| OTTT | 5 | 512 | $63.5\% \pm 2.1$ | $66.2\% \pm 0.7$ |
| OSTL | 5 | 512 | $63.9\% \pm 1.0$ | $67.2\% \pm 1.0$ |
| BPTT | 5 | 512 | $\mathbf{77.7}\% \pm 0.6$ | N/A |
| OTPE | 5 | 512 | $76.4\% \pm 1.2$ | $\mathbf{76.7}\% \pm 0.7$ |
| A. OTPE | 5 | 512 | $74.3\% \pm 0.9$ | $\mathbf{76.7}\% \pm 0.8$ |
| F-OTPE | 5 | 512 | N/A | $74.1\% \pm 1.0$ |
| F-A. OTPE | 5 | 512 | N/A | $72.6\% \pm 1.3$ |

Table 2: Summary of test accuracy on SHD for different model configurations. Results for online learning report highest accuracy determined after learning rate search (Appendix A.2).

et al. (2023), where the approximate gradient calculation of Shrestha and Orchard (2018) is more sensitive to the surrogate derivative slope than exact gradient calculation.

Results comparing online learning performance across the different algorithms are shown in Fig. 6. Online training performance on R-Randman is consistent with those observed for offline training, with all approaches delivering similar accuracy (see training loss in Appendix A.3). The performance difference across OTPE, its approximations, OSTL, and OTTT is more apparent for T-Randman and SHD. Although we observed higher test accuracy for online learning on SHD than for offline learning, we attribute this to our hyperparameter search (see Appendix A.2). As seen in Fig. 6 (c), when evaluated on SHD, we observe that both F-OTPE and F-Approx OTPE converge in performance earlier than OTPE. F-OTPE reaches its peak average performance on SHD after 4,500 mini-batches of online training. Meanwhile, OTPE does not appear to have reached peak validation accuracy even after training on 10,000 mini-batches. We also observe a substantial performance benefit of F-OTPE on the Spiking Speech Commands (Cramer et al., 2020) dataset in Appendix A.4.

## 5 DISCUSSION AND CONCLUSIONS

We propose OTPE and its approximations to facilitate efficient online training in SNNs, by capturing temporal effects typically omitted by similar algorithms. While maintaining similar scalability to OSTL in both compute and memory costs, OTPE produces superior results while remaining layer-local. OTPE and its approximation demonstrate greater alignment to exact gradients in the hidden layers, which may be more beneficial in tasks requiring greater network depth. The training trajectories in the loss landscape consistently demonstrate that models trained with OTPE or its optimizations are closer in model weight-space to BPTT models than those trained using OTTT and OSTL. We evaluated our algorithms on SHD and on rate- and temporal- variations of the Randman task (R-Randman and T-Randman). We observe similar performance across all algorithms when evaluated on R-Randman. However, the temporal influence approximations used by OSTL and OTTT result in degraded performance on temporal tasks like T-Randman and SHD, with OTPE and its variants consistently outperforming them across model configurations and datasets.

# 6 REPRODUCIBILITY

The code for our experiments can be found in the attached supplementary materials. The `readme.txt` file outlines the scripts for generating data and plots.

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

## A  APPENDIX

### A.1  RANDMAN AND SHD HYPERPARAMETERS

Our main parameters are listed here. All others are accessible in the code.

Our Randman parameters are as follows (see `generate_randman_data.py` and `randman_dataset.py` in code):

- 3-dimensional manifolds
- 10 manifolds to classify
- alpha = 1.
- neurons = 50
- time-steps = 50
- batch size = 128
- Each batch is randomly sampled from the underlying manifolds

Our SHD parameters are as follows (see `generate_SHD_data.py` in code):

- time-steps = 50 (reduced to 50 by binning)
- neurons = 700
- 10% of the training set is used for validation. The reported test accuracy uses the model parameters with the best validation accuracy, which is evaluated after each training batch.

### A.2  SHD HYPERPARAMETER SWEEPS

We evaluate offline training on SHD across multiple model sizes and fast sigmoid slopes. One goal of this hyperparameter search is to determine which slope and model size yield the best performance. Another is to identify any trends across model size and slope. As seen in Fig. 7, BPTT consistently performs best. With the exception of 128 layer width, BPTT achieves the highest average score with a slope of 25. Interestingly, we see that going from three to five layers increases the performance of shallower slopes. Using five layers, however, appears to result in poorer performance from OTTT and OSTL, though BPTT also marginally drops. OTPE and its approximation appear to benefit from increased layer depth. These results indicate OTPE has better depth scalability than OTTT and OSTL, especially at sharper slopes.

When we only look at the offline performance of the approximate algorithms, we see OTPE achieving the best average accuracy in the entire sweep and does so using the best slope for BPTT. OTPE also achieves the highest average accuracy in most configurations while also having the most consistent performance across all configurations.

### A.3  ADDITIONAL TRAINING PLOTS

We additionally include training curves evaluated over different model configurations and hyperparameters in Figs.8 – 13. While the mean gradient cosine similarity for each layer shows a decrease in cosine similarity in earlier layers, Fig. 10 shows the gradient cosine similarity changing throughout training. Notably, OTTT's cosine similarity increases throughout training in the output layer while both OSTL and OTTT decrease throughout training in the hidden layers. OTPE, on the other hand, increases in the last hidden layer but decreases in earlier hidden layers. In Fig. 12, OTPE decreases in validation accuracy after training on around 5,000 mini-batches. Meanwhile, the loss continues to decrease, which indicates overfitting.

### A.4  SPIKING SPEECH COMMANDS

We additionally evaluate all online algorithms on the Spiking Speech Commands (SSC) dataset (Cramer et al., 2020). This is a spikified version of the Speech Commands dataset (Warden, 2018), where each sample is a 1-second audio clip of a single spoken word. The dataset has 35 classes and contains 105,829 files, and we use 60,000 of these samples for training and 20,382 for testing. As seen in Table 3, F-OTPE outperforms all others by at least 10%. While we did not perform a learning rate search for SSC, we tested two learning rates. Similar to the results of the hyperparameter sweep, F-OTPE performs poorly with a learning rate of 0.001. Decreasing the learning rate, improved performance for all algorithms. As seen in Fig. 14, the models show slowed improvement by 2,000 mini-batches into training. While the models may continue to improve slowly, there is no indication that OTTT or OSTL will surpass OTPE or F-OTPE in performance.

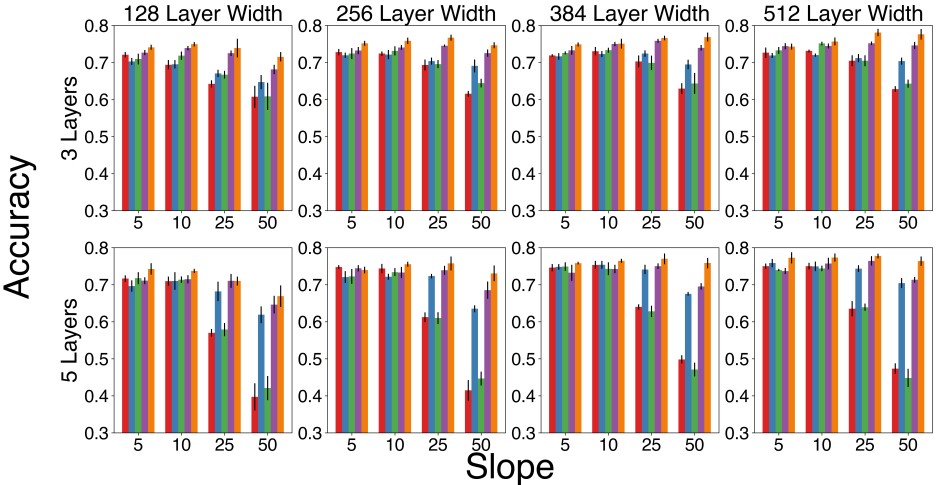

Figure 7: Accuracy results during hyperparameter search.

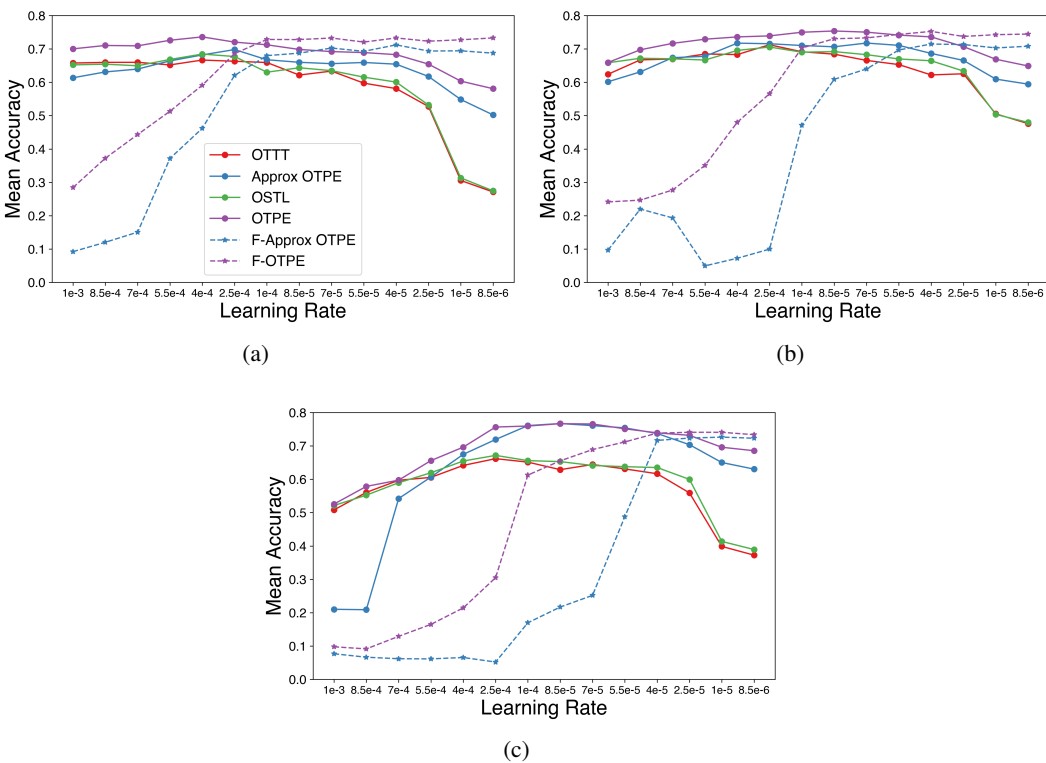

Figure 8: Learning rate hyperparameter search for online learning on SHD. Each point is the mean accuracy across 4 seeds. Figures (a), (b), and (c) are model configurations 3 layers with 128 layer width, 3 layers with 512 layer width, and 5 layers with 512 layer width, respectively.

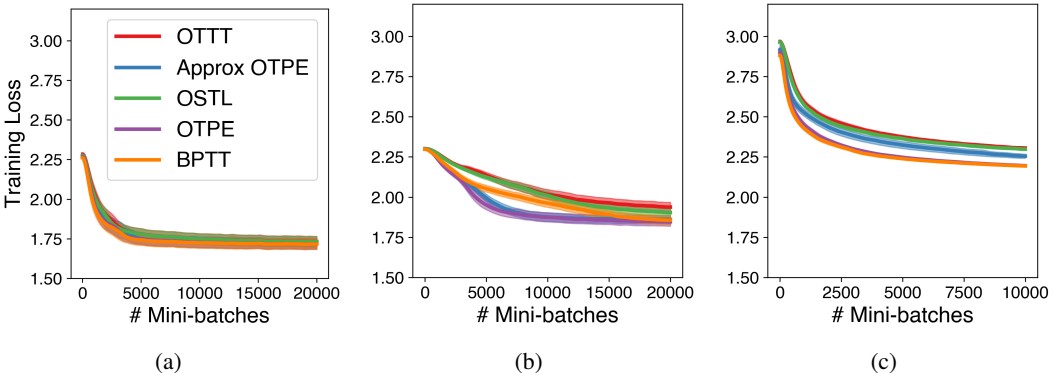

Figure 9: Mean training loss across four seeds throughout offline training. Figures (a), (b), and (c) are R-Randman, T-Randman, and SHD, respectively. Shaded regions are one standard deviation.

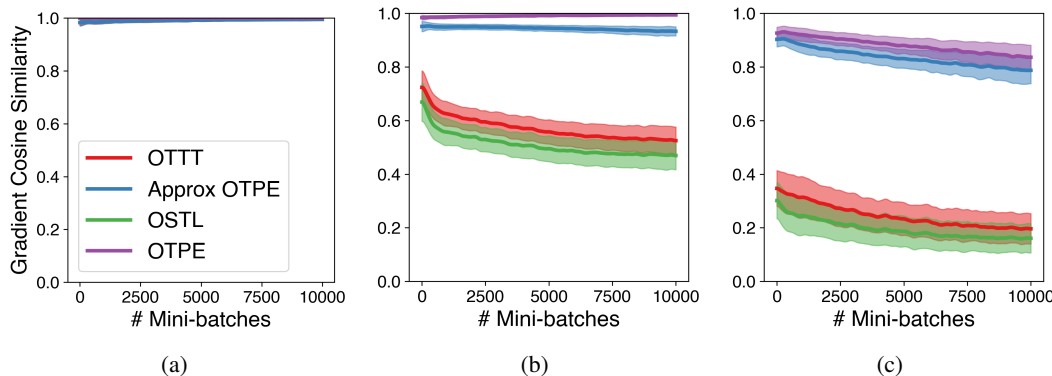

Figure 10: Mean layer-wise gradient cosine similarity across four seeds throughout offline training on SHD. Figures (a), (b), and (c) are the output layer, second hidden layer, and first hidden layer of 512 layer width, respectively.

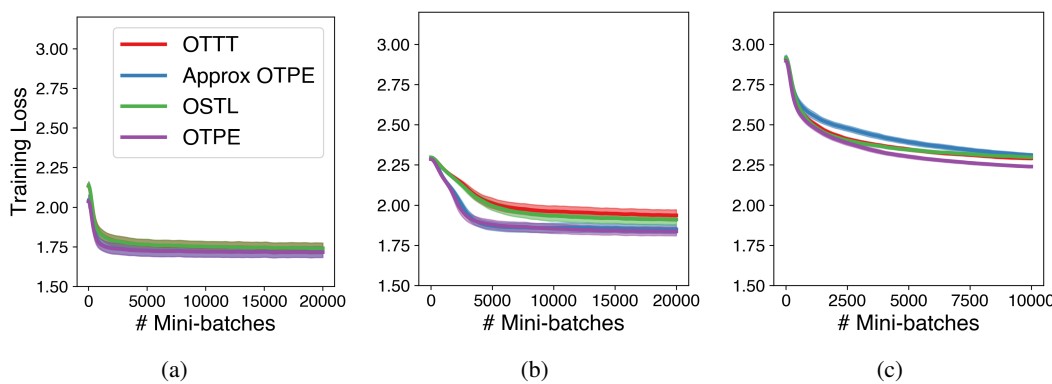

Figure 11: Mean training loss across four seeds throughout online training. Figures (a), (b), and (c) are R-Randman, T-Randman, and SHD, respectively. Shaded regions are one standard deviation.

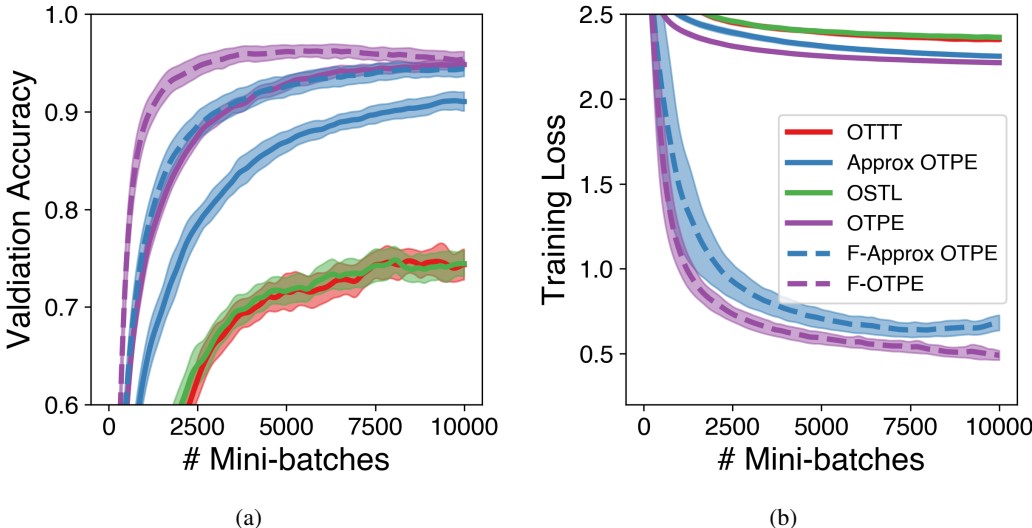

(a)                                                    (b)

Figure 12: Mean training loss and validation accuracy across four seeds throughout online training on SHD. In (b), F-OTPE and F-Approx OTPE calculate loss on the leaked accumulation of the network's output, causing a large difference in the loss value. Shaded regions are one standard deviation.

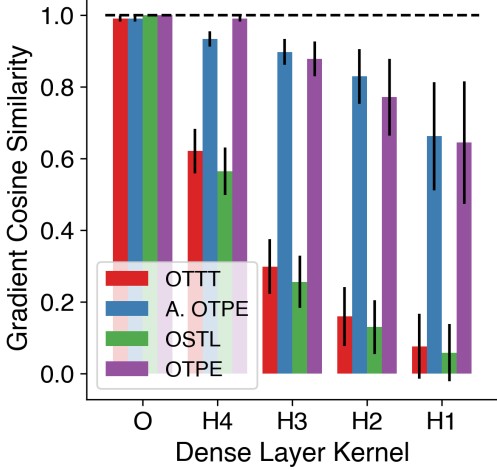

Figure 13: Gradient cosine similarity for a 5-layer model evaluated on SHD. The error bars are one standard deviation across four seeds and all 10,000 measurements throughout training. In the first hidden layer, gradient cosine similarity is sometimes negative for OSTL and OTTT, which is reflected in the error bars spanning below zero.

| Name | Depth | Width | lr=1e$-$3 | lr=1e$-$4 |
|------|-------|-------|-----------|-----------|
| OTTT | 3 | 128 | $18.5\% \pm 0.7$ | $20.0\% \pm 0.6$ |
| OSTL | 3 | 128 | $17.8\% \pm 0.4$ | $21.2\% \pm 0.4$ |
| OTPE | 3 | 128 | $19.9\% \pm 0.7$ | $24.2\% \pm 0.4$ |
| A. OTPE | 3 | 128 | $19.1\% \pm 0.7$ | $19.7\% \pm 0.9$ |
| F-OTPE | 3 | 128 | $6.5\% \pm 0.6$ | $36.1\% \pm 0.8$ |
| F-A. OTPE | 3 | 128 | $6.1\% \pm 0.4$ | $26.1\% \pm 1.0$ |

Table 3: Summary of test accuracy on SSC across 4 seeds. All models were trained online for 5,000 mini-batches. The setup and hyperparameters are similar to SHD.

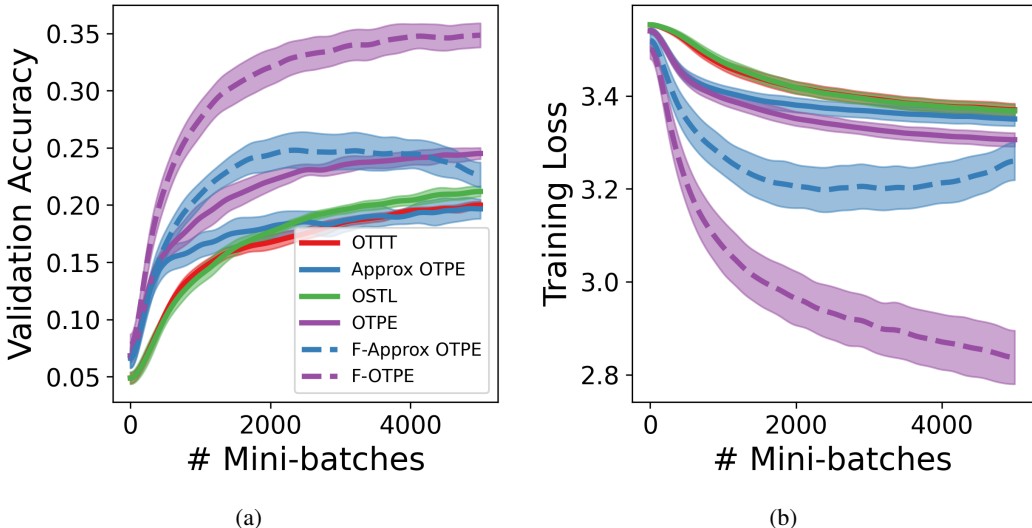

(a)  (b)

Figure 14: Mean training loss and validation accuracy across four seeds throughout online training on SSC for 5,000 mini-batches with a learning rate of $1e-4$. In (b), F-OTPE and F-Approx OTPE calculate loss on the leaked accumulation of the network's output, causing a large difference in the loss value. Shaded regions are one standard deviation.

