# OpenReview forum: "Estimating Post-Synaptic Effects for Online Training of Feed-Forward SNNs"
_ICLR.cc/2024/Conference — Submitted to ICLR 2024_

### Official Review · Reviewer_Rp8L · 2023-10-30

**Soundness:** 2 fair
**Presentation:** 2 fair
**Contribution:** 3 good
**Rating:** 5
**Confidence:** 4

**Summary:**

This paper proposes a new online training method OTPE for spiking neural networks. It incorporates temporal dynamics not captured by existing online methods such as OSTL and OTTT while maintaining similar time and space complexity. Experiments on synthetic datasets and SHD with feedforward networks demonstrate improved performance and better alignment of gradient angles with BPTT.

**Strengths:**

1. This paper considers facilitating online learning in SNNs, which is important for biological plausibility and neuromorphic hardware.

2. Experiments demonstrate large improvements on gradient angle alignment and better performance than existing methods.

**Weaknesses:**

1. The presentation can be improved. It is not quite clear what is the major component in the methodology that leads to the large improvement over existing methods. From Section 3, OTPE is very similar to OSTL and there is no explicit description of the key difference. From Section 3.1, AOTPE differs from OTTT only in that AOTPE uses a running weighted average of surrogate derivatives. Why this small change can almost double the cosine similarity given the similar time and space complexity? Is there more formal theoretical justification? From the derivation, OTPE also makes many approximations, why these approximations have less effect than OSTL/OTTT? Besides, what does the term “postsynaptic estimates” mean?

2. Scalability is mentioned multiple times in the paper. Are there more large-scale (datasets, network, etc.) results?

3. In the abstract, “rate-based and time-based encoding” is mentioned. However, there is no corresponding part in the paper.

**Questions:**

See weakness.

---

> ### Author Response · Authors · 2023-11-23
>
> Dear reviewer,
>
> Thank you for your comments and questions. Our responses are below.
>
> > The presentation can be improved. It is not quite clear what is the major component in the methodology that leads to the large improvement over existing methods. From Section 3, OTPE is very similar to OSTL and there is no explicit description of the key difference. From Section 3.1, AOTPE differs from OTTT only in that AOTPE uses a running weighted average of surrogate derivatives. Why this small change can almost double the cosine similarity given the similar time and space complexity? Is there more formal theoretical justification? From the derivation, OTPE also makes many approximations, why these approximations have less effect than OSTL/OTTT? Besides, what does the term “postsynaptic estimates” mean?
>
> The major component is including the effect of previous spikes, captured by $\hat{R}$. For approx. OTPE, it is the combination of both $\bar{g}$ and $\hat{z}$. The formal justification can be seen in what OSTL misses ($\sum\_{t=1}^T \frac{\partial \mathcal{L}\_t}{\partial s^l\_t} \frac{\partial s^l\_t}{\partial U^l\_t} \left(\frac{\partial U^l\_t}{\partial \theta^{l-1}\_t}+\xcancel{\frac{\partial U^l\_t}{\partial U^l\_{t-1}} \frac{\partial U^l\_{t-1}}{\partial \theta^{l-1}}}\right)$) and how OTPE approximately captures that ($\xcancel{\frac{\partial U^l\_t}{\partial U^l\_{t-1}} \frac{\partial U^l\_{t-1}}{\partial \theta^{l-1}}} \approx \lambda \cdot {\theta^{l}}^\intercal \hat{R}^{l-1}$). Intuitively, hidden layers early in the pipeline of a very long network will affect many membrane potentials, which influences the network outputs in subsequent time-steps. Gradients from this path are ignored by OTTT and OSTL, and as seen by layer-wise gradient cosine similarity, those effects can be the primary driver of gradient direction in early hidden layers.
>
> OSTL’s approximation of gradients is the assumption that the spiking activity at the most recent time-step is solely responsible for the network output at said time-step. OTPE’s approximation directly addresses an area of gradient calculation where OSTL is almost guaranteed to be wrong. As long as OTPE’s approximation is more accurate than assuming this part of the gradient is zero, OTPE’s gradients will be more accurate than OSTL.
>
> “Post-synaptic estimates” refers to $\hat{R}$, which estimates how a layer’s parameters influenced the following layer’s (post-synaptic neurons) membrane potentials.
>
> > Scalability is mentioned multiple times in the paper. Are there more large-scale (datasets, network, etc.) results?
>
> We consider our algorithm scalable because its storage is $O(n^2)$ (i.e. the number of parameters in the network) compared to RTRL with $O(n^3)$ storage (number of parameters x number of state variables), with a similar decrease in computational complexity. We do not provide an empirical comparison because evaluating RTRL is impractical for the multilayer models we are testing.
>
> > In the abstract, “rate-based and time-based encoding” is mentioned. However, there is no corresponding part in the paper.
>
> We introduce R-Randman which is a variant of the Randman dataset that depends only on rate-based encoding in spiking neurons. We then evaluate the performance difference of different algorithms on R-Randman and T-Randman (the time-encoded version of Randman). The rate-based and time-based encoding in the abstract reference these evaluations. The loss landscapes we presented (see Fig 5 ) are for the different learning methods on R-Randman and T-Randman. Figures 2-6 all have subplots for R-Randman and T-Randman for comparison.

---

### Official Review · Reviewer_Q7D9 · 2023-10-31

**Soundness:** 3 good
**Presentation:** 2 fair
**Contribution:** 2 fair
**Rating:** 3
**Confidence:** 4

**Summary:**

In this paper, the author presents an Online Training algorithm with postsynaptic estimates (OTPE). OTPE leverages the preservation of multiple temporal spike outputs to achieve a more precise gradient. Their experiments demonstrate that, compared to similar algorithms, OTPE achieves better alignment with gradients by BPTT.

**Strengths:**

The authors introduce a new approximation RTRL algorithm called OTPE that captures richer temporal effects compared to previous online training methods. OTPE algorithm achieve  gradients highly aligned between BPTT method and effectively reduce the time and space complexity.

**Weaknesses:**

1. The OTTT algorithm was experimented on several datasets such as CIFAR10 and CIFAR100. In comparison, the paper only presented comparative experiments on SHD, which may lead to insufficient persuasive power of the experiments. Therefore, it may be necessary to conduct experiments on a more diverse range of datasets to fully validate the effectiveness of OTPE.
2. The presentation of the Approximate OTPE section in the paper seems unclear as it does not clearly demonstrate the mathematical approximation made by OTPE and Approximate OTPE.
3. The paper claims in the abstract that "This approximation incurs minimal overhead in the time and space complexity compared to similar algorithms." However, the theoretical analysis does not demonstrate the advantages, and there is a lack of corresponding experimental comparisons to support this claim.
4. The LIF neurons of the article seem to be missing $ s_t^{l-1}\cdot \theta $, so the membrane potential stays the state of decay. If the membrane potential update formula is modified to $s_t^l=H(U_{t}^l-V_{th}), U_t^l=\lambda U_{t-1}^l+ s_t^{l-1}\cdot \theta -V_{th}\cdot s_t^l$, then the OTPE's $ \frac{\partial U_t^l}{\partial \theta^{l-1}}=\frac{\partial U_t^l}{\partial s_t^{l-1}}\frac{\partial s_t^{l-1}}{\partial \theta^{l-1}} $ of the second term on the right-hand side What does it mean? Does this also mean that it is possible to compute the expression of the left form directly, without the need to use the chain rule, as in the article?
5. In the second paragraph of section 3, the expression $$\frac{\partial s(t)_i^l}{w_{ij}^l} $$, does the subscript "i" in the numerator refer to the current time step? Could you explain the meanings of the variables and improve the wording accordingly? Moreover, should the parameter "w" be denoted as $\theta$ based on the preceding text?

**Questions:**

See the weakness.

---

> ### Author Response · Authors · 2023-11-23
> **Response to Questions**
>
> Dear reviewer,
>
> Thank you for your comments and questions. Our responses are below.
>
> > W1
>
> We appreciate the reviewer's concerns. However, as we show in Figure 2a, we do not see any advantage to using our method in datasets with limited temporal dependency (e.g., rate-based encoding or the use of datasets like DVS-CIFAR or DVS-Gesture, etc.). We see performance advantages for temporal dependencies in Figure 2b, so we focus on benchmarks for this purpose (e.g. Spiking Heidelberg Digits). We have also added results for Spiking Speech Commands. The relative performance of the online algorithms on SSC is consistent with our other results in online learning.
>
> > W2 and W3
>
> OTPE differs from OSTL by approximating the term OSTL ignores ($\sum\_{t=1}^T \frac{\partial \mathcal{L}\_t}{\partial s^l\_t} \frac{\partial s^l\_t}{\partial U^l\_t} \left(\frac{\partial U^l\_t}{\partial \theta^{l-1}\_t}+\xcancel{\frac{\partial U^l\_t}{\partial U^l\_{t-1}} \frac{\partial U^l\_{t-1}}{\partial \theta^{l-1}}}\right)$), which is the temporal gradient from post-synaptic neurons. Because $ \hat{R}\_{t=0} = \frac{\partial U^l_{t=0}}{\partial \theta^{l-1}_{t=0}} $ and updates as $\hat{R}\_t = \frac{\partial U^l\_t}{\partial \theta^{l-1}\_t}+ \lambda \cdot \hat{R}\_{t-1}$, $\hat{R}$ is the same shape as $\frac{\partial U^l_t}{\partial \theta^{l-1}_t}$, which is $n^2$ due to OSTL's sparse assumption.
>
> $\xcancel{\frac{\partial U^l_t}{\partial U^l_{t-1}} \frac{\partial U^l_{t-1}}{\partial \theta^{l-1}}} \approx \lambda \cdot {\theta^{l}}^\intercal \hat{R}^{l-1}$
>
> Approximate OTPE relies on two assumptions on top of OTPE: 1) We decouple the components of $\hat{R}$, $\frac{\partial s^l_t}{\partial U^l_t}$ and $\frac{\partial U^l_t}{\partial \theta^l}$, through time by assuming $\sum_{t=1}^T \lambda^{T-t} \frac{\partial s^l_t}{\partial U^l_t}\frac{\partial U^l_t}{\partial \theta^l} \approx \left(\frac{1}{T} \sum_{t=1}^T \lambda^{T-t} \frac{\partial s^l_t}{\partial U^l_t} \right) \cdot \left(\sum_{t=1}^T \frac{\partial U^l_t}{\partial \theta^l} \right)$. We maintain a size $n$ running weighted average of the surrogate gradients through time, which we refer to as $\bar{g} = \frac{1}{T}\sum_{t=1}^T \lambda^{T-t} \frac{\partial s^l_t}{\partial U^l_t}$. 2) We approximate our layer-local temporal calculation in OTPE to only store vectors of size n by assuming the same temporal dynamics as OTTT for a single layer, $\frac{\partial U^l_{t}}{\partial \theta^{l}_t} \approx \hat{a}$ , such that $\frac{\partial U^l}{\partial \theta^{l-1}} \approx {\theta^{l}}^\intercal \cdot \bar{g} \left(\hat{a}^{l-1}\_t + \lambda \cdot \hat{z}^{l-1}\_{t-1} \right)$, where $\hat{z}$ is a weighted sum of OTTT's weighted sum ($\hat{z}^{l-1}_t = \hat{a}^{l-1}\_t + \lambda \cdot \hat{z}^{l-1}\_{t-1}$).
>
> > W4
>
> We appreciate the reviewer’s concern. The equations in question specifically address the reset mechanism. We have restructured the equations and text to clarify this.
>
> Yes, updating $\hat{R}$ still does not require the chain rule. Because we assume the weights do not change throughout time when calculating gradients, matrix multiplication with the kernel at each time-step is linear throughout time (i.e. $\sum\_{t=1}^T \theta^{l} s^{l-1}\_t = \theta^{l} \sum\_{t=1}^T s^{l-1}\_t$). Therefore, the involvement of $\theta^l$ in gradient calculation is handled outside of the time loop. If we assume the output layer never spikes, which produces the same gradients as ignoring gradients originating from the reset mechanism, then $U = \theta^{l} \sum\_{t=1}^T \lambda^{T-t}  s^{l-1}\_t$. It follows that $\frac{\partial U^l\_{t}}{\partial \theta^{l-1}} = \theta^{l} \sum_{t=1}^T \lambda^{T-t}  \frac{\partial s^{l-1}\_t}{\theta^{l-1}}$. Since only items within the summation must be calculated through time, we only need the global leak, $\lambda$, and $\frac{\partial s^{l-1}t}{\theta^{l-1}}$ for updating. This avoids the chain rule because both $\lambda$ and $\frac{\partial s^{l-1}t}{\theta^{l-1}}$ are available at layer l-1.
>
> > W5
>
> Thank you for catching this typo. We have changed the notation to match the other equations

---

### Official Review · Reviewer_hRpc · 2023-10-31

**Soundness:** 2 fair
**Presentation:** 2 fair
**Contribution:** 3 good
**Rating:** 5
**Confidence:** 3

**Summary:**

This paper presents a novel method for the online training of feed-forward Spiking Neural Networks (SNNs) that incorporates temporal information from the previous and current spikes into a postsynaptic estimate of the Real-Time Recurrent Learning (RTRL). Previous implementations of online SNN learning (such as Online Training Through Time (OTTT) or Online Spatio-Temporal Learning (OSTL)) fails to account for the temporal information of previous spikes leading to reduced accuracy of the SNN. Using Online Training with Postsynaptic Estimates (OTPE) or its approximation, gradient estimates exhibited more alignment to the exact gradients generated from Backpropagation through time (BPTT) compared to previous approaches.

**Strengths:**

-	Good background explanation, especially with explaining the mathematics of previous methods in comparison to OTPE.
-	Table 2 clearly demonstrates how OTPE performs at a closer online accuracy level to BPTT offline accuracy for SNNs at different depths and widths compared to other approximation training methods.
-	Analysis was clear and transparent. Analysis was able to differentiate between actual results and faulty results due to the methodology. In the last paragraph before Section 5, the author says “The performance difference across OTPE, its approximations, OSTL, and OTTT is more apparent for T-Randman and SHD. Although we observed higher test accuracy for online learning on SHD than for offline learning, we attribute this to our hyperparameter search (see Appendix A.2).” Author also acknowledges intermediate/incomplete results in this paragraph when it’s said “OTPE does not appear to have reached peak validation accuracy even after training on 10,000 mini-batches.”

**Weaknesses:**

-	The F-OTPE method should’ve been tested in multiple application-specific SNN datasets such as Auditory N-MNIST, DVS-Speech, or Spiking MNIST Audio Dataset. It’s hard to tell whether the results presented for F-OTPE are valid and can be reproducible since the method was only tested on one dataset.
-	Online training of OTPE only occurred on one dataset throughout the paper. If this is supposed to be a preferred method for online training, there should be more tests demonstrating the improved performance of OTPE in online SNN training using a variety of datasets such as Auditory N-MNIST, DVS-Speech, or Spiking MNIST Audio Dataset. The author can also try testing using image classification data (instead of audio) by using N-CARS, DvsGesture, or N-MNIST.

**Questions:**

-	[Introduction, 1st paragraph] The authors say that “However, BPTT is unsuitable for online learning (Kaiser et al., 2020; Bohnstingl et al., 2022)”. Throughout the paper, I see them using gradients from BPTT as the ground truth for their approach. Can you explain this discrepancy?
-	In Figure 2, I observe that the proposed approach is as good as BPTT, then why won’t I just use BPTT? What advantage is the OTPE giving?
-	[Introduction, 2nd paragraph] “To address these limitations ...” All the references are bunched up so it is hard to understand which technique points to which reference.
-	Many of the figures were unclear or hard to read, e.g., all the line plots in Fig 2a are bunched up, for Figure 5 the markers are obfuscating the color of the line.
-	Please explain F-OTPE more thoroughly in the Section 3.2. Elaborate on the line “the loss can be calcualted similarly to how cross-entropy loss is calculated in offline training.” Also include the equations involved for F-OTPE and how it relates to normal OTPE. While the explanation in the paper was fine, it will be better explained through a visual of the mathematics.
-	Why wasn’t F-OTPE tested along with OTPE and approx OTPE in the offline training scenarios presented in Figures 2, 3, 4, 5?
-	How will OTPE, approx OTPE, F-OTPE perform in a different application-specific dataset? You can stick to audio datasets by using Auditory N-MNIST, DVS-Speech, Spiking MNIST Audio Dataset, etc. Or you can expand to image classification datasets by using N-CARS, DvsGesture, N-MNIST, etc.

---

> ### Author Response · Authors · 2023-11-23
> **Response to Questions**
>
> Dear reviewer,
>
> Thank you for your comments and questions. Our responses are below.
>
> > [Introduction, 1st paragraph] The authors say that “However, BPTT is unsuitable for online learning (Kaiser et al., 2020; Bohnstingl et al., 2022)”. Throughout the paper, I see them using gradients from BPTT as the ground truth for their approach. Can you explain this discrepancy?
>
> We only compare against BPTT for offline learning scenarios, for online learning we compare against OTTT and OSTL. Like us, they compare with offline learning to evaluate the performance cost of the approximation. This is because online learning with exact gradients (via RTRL) does not facilitate practical evaluation for multi-layer networks and is too costly to use for comparison.
>
> > In Figure 2, I observe that the proposed approach is as good as BPTT, then why won’t I just use BPTT? What advantage is the OTPE giving?
>
> While BPTT can deliver high performance in the offline learning scenario, it is not suitable for online learning (Rostami et al., 2022). In order to contextualize OTPE’s online learning performance, we also compare its offline learning capabilities to BPTT. However, we primarily focus on online learning.
>
> > [Introduction, 2nd paragraph] “To address these limitations ...” All the references are bunched up so it is hard to understand which technique points to which reference.
>
> We have addressed this by placing the citations with their corresponding concept.
>
> > Many of the figures were unclear or hard to read, e.g., all the line plots in Fig 2a are bunched up, for Figure 5 the markers are obfuscating the color of the line.
>
> We addressed this by reducing clutter in Fig 2a and subsampling our points on Figure 5. Unfortunately, space limitations prevent larger figures.
>
> > Please explain F-OTPE more thoroughly in the Section 3.2. Elaborate on the line “the loss can be calcualted similarly to how cross-entropy loss is calculated in offline training.” Also include the equations involved for F-OTPE and how it relates to normal OTPE. While the explanation in the paper was fine, it will be better explained through a visual of the mathematics.
>
> OTPE calculates gradients under the assumption that output spikes of a hidden layer at previous time-steps are accumulated with membrane leak in the following layer ($\hat{R}^{l} = \frac{\partial \sum^{T}\_{t=1} \lambda^{T-t} s^{l}\_t}{\partial \theta^l}$). In the output layer, this can apply to the leaked accumulation of spikes through time. Suppose we calculate loss using the accumulated spikes instead of solely relying on the spiking behavior at the current time-step. In this case, we can use $\hat{R}$ to exactly calculate the derivative of the loss with respect to the output layer’s parameters ($\frac{\partial \mathcal{L}}{\partial \sum^{T}\_{t=1} \lambda^{T-t} s^{l}\_t} \frac{\partial \sum^{T}_{t=1} \lambda^{T-t} s^{l}\_t}{\partial \theta^l} = \frac{\partial \mathcal{L}}{\partial \sum^{T}\_{t=1} \lambda^{T-t} s^{l}\_t} \hat{R}^l = \frac{\partial \mathcal{L}}{\partial \theta^l}$). We have added this text and the mathematics for F-OTPE to the paper.
>
>
> > Why wasn’t F-OTPE tested along with OTPE and approx OTPE in the offline training scenarios presented in Figures 2, 3, 4, 5?
>
> F-OTPE uses a different scoring mechanism than OTTT and OSTL, which prevents a fair comparison of gradient approximation quality and training trajectory. Both OTTT and OSTL are derived under the assumption that the loss only applies to the output of the latest time-step. If you were to use OSTL and score an aggregation of previous model outputs, then the output layer of OSTL would not match exact gradients. The primary purpose of our offline results is to make relative comparisons concerning approximation quality and optimization. We have added text to the manuscript to clarify the exclusion of F-OTPE.
>
>
> > How will OTPE, approx OTPE, F-OTPE perform in a different application-specific dataset? You can stick to audio datasets by using Auditory N-MNIST, DVS-Speech, Spiking MNIST Audio Dataset, etc. Or you can expand to image classification datasets by using N-CARS, DvsGesture, N-MNIST, etc.
>
> We have examined test accuracy after online training on an additional dataset, Spiking Speech Commands. Our results are consistent with our previous results in online learning. F-OTPE, in particular, outperforms all other online algorithms by at least 10%.

---

### Meta-Review · Area_Chair_7UP1 · 2023-12-10

**Metareview:**

The paper introduces Online Training with Postsynaptic Estimates (OTPE), a method for the online training of feed forward SNNs. The authors claim that OTPE effectively incorporates multi-timestep post-synaptic effects into gradient approximation, improving temporal influence calculations. This method provides a closer alignment to exact gradients generated by Backpropagation through time. The authors demonstrate OTPE's effectiveness through experiments, showing its performance in various SNN model configurations and in both rate-based and time-based encoding scenarios.

Strengths:
- The paper provides a clear and detailed background, making a strong case for the need for improved methods in online training of SNNs.
- The paper effectively demonstrates the practicality of OTPE, especially in the context of online learning where computational efficiency is paramount.

Weaknesses:
- The reviewers noted a lack of diverse dataset testing. The paper primarily focuses on a single dataset, which may not adequately represent the method's effectiveness across different applications.
- There are concerns about the clarity and presentation of some sections, particularly regarding the mathematical underpinnings and explanations of OTPE and its variants.
- The paper could benefit from a more extensive theoretical analysis to support its claims, especially in comparison to existing methods like OTTT and OSTL.

Based on reviewers' feedback while taking into account the authors' rebuttal, the AC leans toward rejecting the paper. The AC also encourages the authors to improve the paper based on the feedback they received from the reviewers.

**Justification For Why Not Higher Score:**

The paper needs significant revisions to be prepared for publication as denoted by the reviewers and summarized in the weaknesses points mentioned by the AC.

**Justification For Why Not Lower Score:**

N/A

---

### Decision · Program_Chairs · 2024-01-16

Reject